# Trends in Avoidable Hospitalizations for Heart Failure in Switzerland (1998–2018): A Cross-Sectional Analysis [note 1]

**DOI:** 10.3390/healthcare12242547

**Published:** 2024-12-17

**Authors:** Lionel Gaillard, Ko Ko Maung, Charlène Mauron, Pedro Marques-Vidal, Alexandre Gouveia

**Affiliations:** 1Centre for Primary Care and Public Health (Unisanté), University of Lausanne, 1011 Lausanne, Switzerland; lionel.gaillard@bluewin.ch (L.G.); charlene.mauron@chuv.ch (C.M.); 2Department of Medicine, Internal Medicine, Lausanne University Hospital (CHUV), University of Lausanne, 1011 Lausanne, Switzerland; ko-ko.maung@chuv.ch (K.K.M.); pedro-manuel.marques-vidal@chuv.ch (P.M.-V.)

**Keywords:** potentially avoidable hospitalizations, heart failure, administrative data, Switzerland

## Abstract

**Background/Objectives**: Timely and appropriate outpatient care can prevent potentially avoidable hospitalizations (PAHs) for heart failure (HF). We analyzed the trends, determinants, and consequences of PAHs for HF in Switzerland over two decades.; **Methods**: Hospital discharge data of Switzerland from 1998 to 2018 were utilized. PAH was defined according to the Organization for Economic Cooperation and Development (OECD) criteria.; **Results**: Data from 206,000 PAHs for HF were included (49.1% women, 55.8% aged over 80). Admission rates for PAHs represented 54.5 per 10,000 admissions in 1999, and they increased to 117.6 per 10,000 admissions in 2018. Similarly, age-standardized admission rates were 107.8 per 100,000 inhabitants in 1999, and they increased to 220.7 per 100,000 inhabitants in 2018. Between 1999 and 2018, patients admitted with PAHs for HF became older (% of patients aged over 80: 60.4% in 2018 vs. 49.2% in 1999), presented more frequently with a Charlson index < 4 (65% vs. 35%), were admitted more frequently as an emergency (89.0% vs. 60.7%), by the patient’s own initiative (31.5% vs. 13.9%), while ICU admission increased only slightly (8.6% vs. 7.6%) and length of stay decreased—median and (interquartile range) 8 (6–13) vs. 12 (8–18) days. In 2018, the costs related to PAHs for HF were estimated at over CHF 170 million, and the corresponding number of occupied beds at 407 per year; **Conclusions**: In Switzerland, the number of PAHs for HF has increased steadily. The medical and financial burden due to PAH for HF could still be reduced with timely and appropriate outpatient care.

## 1. Introduction

Cardiovascular disease (CVD) is the leading cause of death globally with almost one-third of the total deaths worldwide, i.e., around 18.5 million deaths in 2019 [1]. It is also the major cause of morbidity and disability-adjusted life years around the world. Cardiovascular diseases refer to a group of disorders of the heart and blood vessels such as ischemic heart disease, cerebrovascular disease, heart failure (HF), and peripheral arterial diseases. Among CVDs, HF is the leading cause of hospitalizations in Europe with an approximated 1–2% of all hospitalizations [2]. Consequently, the economic burden associated with the healthcare expenditure of HF is alarming. Indeed, the annual cost due to heart failure hospitalizations is estimated at almost USD 16,000 per patient in the United States [3] and the overall cost worldwide was approximately USD 108 billion in 2012 [4]. Furthermore, HF has been highlighted as a global pandemic with an estimated prevalence of 64.3 million worldwide in 2017 [5]. With the global ageing population and multimorbidity [6], the medical and economic burden associated with HF is projected to increase even more. Nevertheless, HF is one of the ambulatory care sensitive conditions (ACSCs) where better management in primary care could potentially avoid certain hospitalizations [7]. To reduce the burden due to HF, it is crucial to identify the rate, trends, determinants, and cost of hospitalizations that are potentially avoidable. This would consequently help the healthcare stakeholders to improve management and prevention strategies for chronic diseases at the primary care level.

Potentially avoidable hospitalizations (PAHs) are the hospitalizations that could have been avoided if there had been proper and effective outpatient primary care or early interventions outside the hospital settings. They have been widely used as an indicator of the performance, quality, and access to primary healthcare [8]. Indeed, PAHs can be useful to identify and prioritize areas to improve access to healthcare and efficiency of outpatient care, and consequently, to reduce the disease burden. PAHs for other ACSCs such as asthma and chronic obstructive pulmonary disease in Switzerland showed an upward trend in the number of PAHs in 2018 with almost double the number of PAHs for COPD in 2008 [9].

HF is a chronic and impairing disease, the worldwide prevalence of which has more than doubled between 1990 and 2019 [10]. Despite being one of the countries with the best quality healthcare system in the world, there were considerable variations among PAHs of HF in Switzerland in 2017 [11]. However, no studies have been done regarding the trends and determinants of PAH for HF in Switzerland. Therefore, we aimed to analyze the trends, determinants, and consequences of PAH for HF in Switzerland for two decades from 1999 to 2018.

## 2. Materials and Methods

### 2.1. Data Sources

Nationally representative data for Switzerland were obtained from the Swiss Federal Office for Statistics, for the period 1999 to 2018 (contract number 200291). The data cover over 98% of public and private hospitals within Switzerland and include all stays for each hospital [12].

### 2.2. Variables, Data Sources, and Measurements

The main cause for hospitalization and the comorbidities were coded using the 10th revision of the International Classification of Diseases (ICD-10) of the WHO. The data also contain information regarding gender, age (categorized into 5-year groups), administrative regions (26 Swiss cantons), date of admission (limited to month and year), decision of admission (i.e., patient’s or doctor’s initiative, others), type of admission (planned or emergency), type of room (infirmary, semi-private, or private room), admission to intensive care unit (ICU), type of intervention (coded using the Swiss CHOP system) [13], and length of stay (LOS). For this analysis, we grouped the age categories into 10-year age groups and the 26 cantons into seven administrative regions, i.e., Leman, Mittelland, Northwest, Zurich, Eastern, Central, and Ticino.

Severity of disease was assessed using the Charlson comorbidity index adapted to the Swiss population, as described previously [14]. The index was computed using data from the current hospitalization and patients were categorized into <4 and 4+ score values.

### 2.3. Potentially Avoidable Hospitalizations and Their Consequences

The definition for a PAH for HF was obtained according to the international OECD Health Care Quality Indicators Project criteria [13]. The inclusion and exclusion criteria are summarized in Appendix A. Only data from adults, defined as age group 20–24 and upper, were included.

The total number of days due to PAHs was computed for each year. This number was then divided by 365 to obtain the minimum number of beds that would be theoretically solely dedicated to PAHs during that year. A pragmatic value of 85% occupation was used to estimate the upper bound.

Costs were computed for year 2018 using the Swiss Diagnosis-Related Group (DRG) system, as indicated previously [9], and values were expressed in CHF (CHF 1 = EUR 1.03 or USD 1.12 as of 22 July 2024)

### 2.4. Inclusion and Exclusion Criteria

Only data related to adults (i.e., being at least in age group 20–24) were eligible for analysis. Patients coming from outside of Switzerland or hospitalizations with missing covariates were excluded.

### 2.5. Statistical Analysis

Analysis was conducted using Stata version 18.0 (Stata Corp, College Station, TX, USA). Descriptive results were expressed as number of hospitalizations (percentage) or as median (interquartile range).

PAH rates for HF were computed for each calendar year using as denominator either the number of admissions (rate expressed as number of PAHs per 10,000 admissions), or the population of each region aged over 20 as obtained from the Swiss Federal Office of Statistics [15] (rate expressed as PAHs per 100,000 inhabitants). Standardized rates of PAHs for each calendar year were computed via direct standardization using the 2013 standard EU population and expressed per 100,000 inhabitants. Trends in rates were assessed using linear regression with year and year squared as independent variables.

Multivariable analysis of the factors associated with emergency admission or ICU stay were assessed using logistic regression and the results were expressed as odds ratio (OR) and (95% confidence interval). Statistical significance was considered for a two-sided test with *p* < 0.05.

### 2.6. Ethical Statement

The data of the Swiss Federal Office of Statistics are available for research purposes and, therefore, neither specific individual consent nor authorization from an Ethics Committee was needed. Data that are collected routinely in the Medical Statistics of Hospitals are de-identified.

## 3. Results

### 3.1. General Characteristics of Potentially Avoidable Hospitalizations

The initial database gathered a total of 23,952,684 hospital admissions in Switzerland between 1999 and 2018. From this data, 674,334 hospitalizations for HF were obtained, of which 206,000 were considered for rates, and 194,452 were considered for the multivariable analyses, as 11,548 had at least one missing variable.

The main characteristics of PAHs for heart failure are summarized in Table 1. Women and men were evenly distributed, while patients aged 80 years were overrepresented in most admissions. Almost nine out of ten patients were of Swiss nationality, almost all had health insurance, and over eight out of ten admissions took place in an emergency. Also, half of the admissions were referred by a doctor, and one-third of the admitted patients had a Charlson index of 4 or more (Table 1).

### 3.2. Trends of Potentially Avoidable Hospitalizations

The trends in PAHs for HF expressed per 10,000 admissions and age-standardized per 100,000 inhabitants, for each administrative region and the whole of Switzerland, are provided in Figure 1 and Figure 2, respectively. Rates increased steadily, from 54.5 per 10,000 admissions in 1999 to 117.6 per 10,000 admissions in 2018, and from 87.9 per 100,000 inhabitants in 1999 to 215.2 per 100,000 inhabitants in 2018. For the whole of Switzerland, age-standardized trends increased from 107.8 per 100,000 inhabitants in 1999 to 220.7 per 100,000 inhabitants in 2018 (Figure 3). All trends were statistically significant at *p* < 0.001 by linear regression.

Between 1999 and 2018, the gender distribution remained relatively constant, the percentage of admissions with patients aged 80 and over increased from 49.2% to 60.7%, the percentage of Swiss decreased from 87.5% to 84.4%, the percentage of admission due to an emergency increased from 60.7% to 89.0%, the referral by the patient’s initiative increased from 13.9% to 31.5%, and the percentage of patients with a Charlson index over 4 increased from 16.9% to 46.7% (Appendix A). After adjusting for the year, women, older age, being non-Swiss, having a high Charlson index, and referral decision by the patient’s initiative or via ambulance or police had a higher likelihood of being admitted via the emergency ward, while differing associations were found regarding the administrative region (Appendix A).

### 3.3. Consequences of Potentially Avoidable Hospitalizations

The consequences of PAHs are indicated in Table 2. Less than one-tenth of the admissions had an ICU stay, and almost three-quarters of the patients admitted returned home. The median length of stay was 10 days, and in 2018, the estimated cost for one PAH was CHF 9313.

Between 1999 and 2018, the percentage of admissions with an ICU stay increased from 7.6% to 8.6% (*p* > 0.05), the discharges to a medicalized home increased from 9.9% to 15.1% at the expenses for discharges to home, and the length of stay decreased from 12 (8–18) to 8 (6–13) days (*p* < 0.001) (analysis by linear regression, Appendix A). After adjusting for the year, women and older age had a lower likelihood of being admitted to ICU; having a high Charlson index, referral decision by the patient’s initiative or via ambulance or police, and admission via the emergency ward had a higher likelihood of being admitted to ICU, while differing associations were found regarding the administrative region (Appendix A).

The total number of hospital days related to PAH for HF increased from 72,561 in 1999 to 148,390 in 2018 ( Appendix A), corresponding to an estimated number of 407 (minimum) to 478 (upper bound) occupied beds for the whole year, representing 1.8% of acute beds in hospital in 2018 which rose from 0.7% in 1999 (Appendix A). In 2018, the overall costs related to PAH for HF were estimated at over CHF 170 million (CHF 170,352,957).

## 4. Discussion

### 4.1. Trends of Potentially Avoidable Hospitalizations

Our study showed an increase in the PAHs for HF during the last two decades, which is consistent with the increase in HF in the general population. In the western world, HF has a stable incidence of 1–20 per 1000 person-years but an increasing prevalence of 1–3% of the general population, probably due to the aging of the population, the better surviving with ischemic heart disease, the improved treatment of HF, and therefore living longer with it [5]. HF is known to be an important burden for the healthcare system and represents 1–2% of all hospitalizations in Western countries and is the first cause of hospitalization in the population over 65 years old [5].

The increase in PAHs for HF that came out of our study is therefore not surprising, in addition to the confirmation of this trend by the OECD results that show the same increase in Switzerland’s Congestive heart failure hospital admission [16]. Different mechanisms can be considered to try to explain this trend. Mostly, the role of the general practitioners is believed to be linked to the number of PAHs. In fact, a study showed that the prevalence of HF patients visiting a GP was surprisingly low in Switzerland when compared to other countries [17]. This could be explained by the high density of ambulatory cardiologists and the absence of gatekeeping in Switzerland [18]. However, even if these cardiologists ensure the follow-up of HF patients, they might not have the same primary care availability as GPs. Countries with modern healthcare systems that have strong primary care, such as Slovenia and Portugal, have been able to maintain low rates of PAH, although those seem mostly related to the propensity to hospitalize, regional factors, and educational level, rather than to outpatient care performance [19].

Other chronic diseases like asthma and COPD have also shown an increase in PAHs but not quite in a similar proportion to HF. Asthma, for example, showed relatively similar proportions of PAHs over total hospitalizations between 1998 and 2018, whereas COPD nearly doubled during this period [9].

### 4.2. Characteristics of Potentially Avoidable Hospitalizations

Our results also showed a balanced ratio between male and female patients in PAHs. This does not correspond to the much higher proportion of males in the HF prevalence and incidence of HF in the general population. Our study showed that the peak of PAH for HF occurred around the age of 80, which is consistent with other studies [17]. This is higher than among incident HF cases, likely due to a better treatment, which maintains patients alive. Therefore, this relatively high proportion of patients aged 80 years old in PAHs for HF is not surprising.

Our data also showed that 80% of PAH patients were admitted in an emergency. This could be an indicator of a potential lack of primary care coverage leading to poorer control of the disease and therefore inducing emergency situations and a propensity to hospitalize. Slightly over half of all admissions were on medical indication. This suggests that patients with HF might not be able to contact their usual carer in due time, and that hospital admission is decided as a replacement for a consultation. Future studies should focus on the reasons for a carer to suggest a PAH for a patient with HF.

The Charlson index is an indicator of the comorbidities of patients admitted to the hospital. MI and other cardiovascular diseases even including HF are listed among other comorbidities as items of this score [20]. Comorbidities are also a complicating factor and might therefore induce higher hospitalization rates, both potentially avoidable and not avoidable [5].

### 4.3. Consequences of Potentially Avoidable Hospitalizations

Due to the high and increasing prevalence of HF, the economic burden of the disease is important and has become a public health issue. Studies showed that in most Western countries, a major part of HF-linked costs are due to the hospitalizations that the disease induces (before medication, rehabilitation, diagnosis, consultation, emergencies, and long-term care) [5]. Our results, showing this important and increasing portion of PAHs among all hospitalizations, are therefore extremely relevant and require particular attention. More than the number of PAHs, the lengths of stay also have shown an increase and allow to calculate the beds claimed for HF PAH in comparison to other diseases, especially non-avoidable hospitalizations. The total sum of days for PAH stays has increased from 72,561 days in 1998 to 148,390 days in 2018, which represents a jump from a range of 199 to 234 beds in 1998 to 407 to 478 beds theoretically solely dedicated to HF PAH 20 years later.

Compared to other studied PAH causes, HF shows a much higher increase in length of stay and number of beds for PAH. In fact, PAH for asthma even showed a decrease in both the number of days and beds between 1998 and 2018, whereas COPD was subject to a threefold increase for both parameters [9].

### 4.4. Implications for Public Health and Clinical Practice

PAHs generally pose a significant economic burden and reduce the quality and efficiency of the healthcare system. Thus, reducing the number of PAHs could significantly enhance healthcare efficiency while also reducing the economic burden. Therefore, multidisciplinary strategies for public health and clinical practice are highly required. In terms of public health education, the European society of cardiology recommends education programs to patients with HF to identify the early symptoms of HF and to adopt good practices such as alternate daily weight monitoring and timely contacting their GP in the case of sudden body weight increases or in the presence of early symptoms of HF [21]. As most PAHs were decided by doctors, it is also crucial to implement continuous medical education on the guidelines for the management of HF [22]. Other strategies such as fast track systems and improving primary care access for acute hospitalizations of HF, especially for the vulnerable and older patients, might also improve the overall efficiency of the healthcare system

### 4.5. Study Strengths and Limitations

Our study has some significant strengths. First, the results that we present are derived from official data from the Office of Federal Statistics that covers the entire country, with hospitalizations that occurred in the entire Swiss healthcare system during a total period of 20 years. Second, the criteria that we used for PAHs are internationally validated and are frequently used to compare the quality of primary care among countries. Thirdly, our study provides additional information about the care that was provided in PAHs for heart failure, such as the admission to the intensive care unit during the hospital stay, and the related costs for PAHs for heart failure. This information is useful for patients, healthcare providers, researchers, and stakeholders, to acknowledge even further the impact of PAHs on the cost and burden of healthcare systems.

This study has several limitations that must be disclosed. First, our study is observational and, therefore, causality cannot be established between the observed trends and the specific factors investigated. Second, the hospitalization data that were used from the Swiss Federal Office for Statistics, although comprehensive, may present inaccuracies or inconsistencies due to coding errors or omissions in hospital records that can be heterogeneous from one region to another. Third, while the study covers a substantial time period, it does not account for potential changes in healthcare policies, coding practices or reimbursement schemes that may have influenced hospitalization rates over the two decades. For example, increasing or decreasing use of HF code to modify reimbursements would artificially change the prevalence of the disease and modify the trends. Unfortunately, no information regarding any change in coding procedures could be found. Fourth, the exclusion of hospitalizations with missing covariates may introduce a selection bias, as only admissions with complete data (and possibly better management) were utilized, potentially impacting the generalizability of the findings. Finally, the study focuses exclusively on Switzerland, and the results may not be applicable to other countries with different healthcare systems, demographics, or health policies.

## 5. Conclusions

In Switzerland, PAHs for heart failure have steadily increased between 1998 and 2018, in all regions of the country. Patients with PAHs for heart failure are mostly over the age of 80 years old, with fewer comorbidities and admitted as emergency hospitalizations. The burden of PAHs is highly significant, with important hospital resources that are mobilized, and healthcare policies such as patient education programs, enhanced primary care access, and implication of updated guidelines to medical professionals are required to introduce effective evidence-based health measures to reverse this trend, at the regional and national levels.

## Figures and Tables

**Figure 1 healthcare-12-02547-f001:**
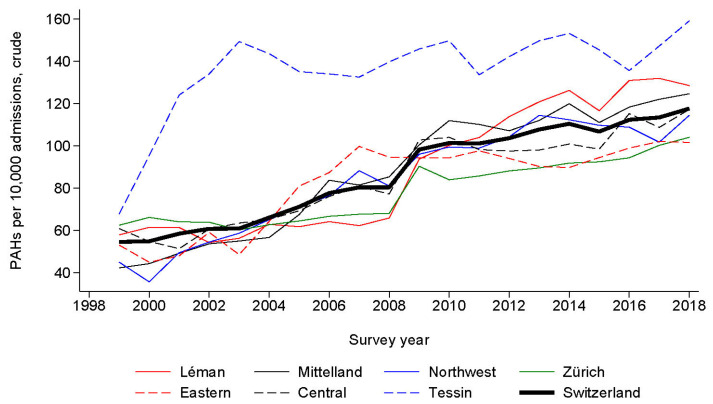
Trends in potentially avoidable admissions for heart failure in Switzerland, overall and by administrative region, for period 1999–2018. Results are expressed as number of potentially avoidable admissions per 10,000 admissions. Except for Tessin, where a considerable increase in PAH occurred in 2000–2004 to level afterwards, all administrative regions show a regular increase in PAH rates.

**Figure 2 healthcare-12-02547-f002:**
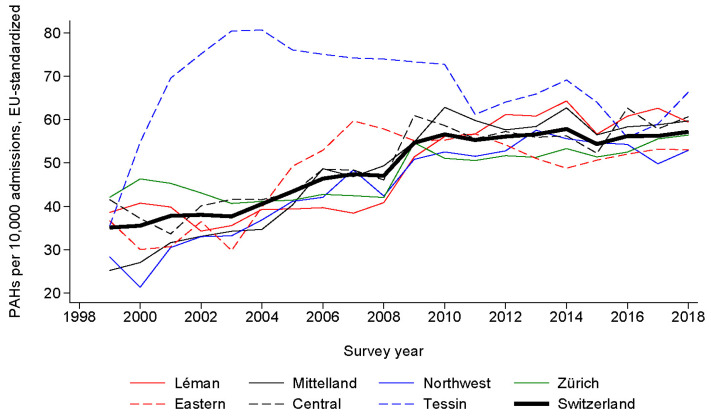
Trends in potentially avoidable admissions for heart failure in Switzerland, overall and by administrative region, for period 1999–2018. Results are expressed as age-standardized rates per 10,000 admissions. Except for Tessin, where a considerable increase in PAH occurred in 2000–2004 to level afterwards, all administrative regions show a regular increase in PAH rates. The rates are lower than in Figure 1, as most admissions occur in elderly people, and standardizing for an entire population would decrease the rate.

**Figure 3 healthcare-12-02547-f003:**
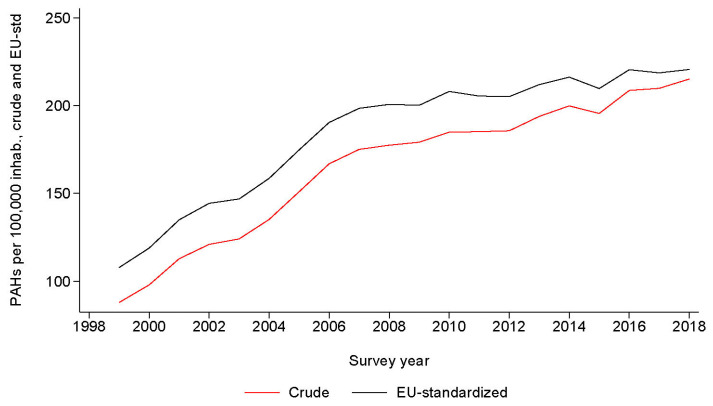
Trends in potentially avoidable admissions for heart failure for Switzerland, for period 1999–2018. Results are expressed as crude and age-standardized rates per 100,000 inhabitants. The two curves show a similar trend, and tend to join in 2018, possibly due to an age distribution of the Swiss population closing to the one of the EU.

**Table 1 healthcare-12-02547-t001:** Characteristics of potentially avoidable hospitalizations (PAHs) for heart failure, Switzerland, 2009–2018.

	Result
Sample size	206,000
Women	101,183 (49.1)
Age groups	
(20–30)	248 (0.1)
(30–40)	703 (0.3)
(40–50)	2502 (1.2)
(50–60)	8533 (4.1)
(60–70)	23,007 (11.2)
(70–80)	56,097 (27.2)
(80–90)	88,424 (42.9)
(90+)	26,486 (12.9)
Region	
Leman	40,751 (19.8)
Mittelland	41,554 (20.2)
Northwest	34,759 (16.9)
Zurich	33,001 (16.0)
Eastern	25,707 (12.5)
Central	14,386 (7.0)
Tessin	15,842 (7.7)
Swiss	180,994 (87.9)
No insurance	5659 (2.8)
Emergency	168,875 (82.0)
Admission	
Patient	46,680 (22.7)
Ambulance	39,830 (19.3)
Doctor	112,031 (54.4)
Other	7459 (3.6)
Charlson index	
<4	133,893 (65.0)
4+	72,107 (35.0)

Results are expressed as number of admissions (column percentage).

**Table 2 healthcare-12-02547-t002:** Consequences of potentially avoidable hospitalizations for heart failure, Switzerland, 1999–2018.

	Result
N	206,000
Type of hospital stay (%)	
Infirmary	161,847 (78.6)
Semi-private	29,392 (14.3)
Private	14,702 (7.1)
Intensive care unit stay (%)	17,855 (8.7)
Destination at discharge (%)	
Home	148,576 (72.1)
Medical home	25,493 (12.4)
Other	31,931 (15.5)
Length of stay (days)	10 (7–15)
Estimated cost (CHF) §	9313 (8717–13,866)

§ for 2018 only. Results are expressed as median (interquartile range) or as number of admissions (column %).

## Data Availability

Due to legal constraints, sharing of the data is not allowed. People interested in obtaining the data should contact the Swiss Federal Office of Statistics (www.bfs.admin.ch) for further information.

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
