# Peer review of "Trends in Avoidable Hospitalizations for Heart Failure in Switzerland (1998–2018): A Cross-Sectional Analysis†"

_healthcare, 2024, doi:10.3390/healthcare12242547_

Round 1

Reviewer 1 Report

Comments and Suggestions for Authors

The article is within the Journal aim and the topic is relevant. The text is clear and methods well described. 

abstract: line 16 2009 to be revised as 1998

line 123-125 please remove

Discussion: Probably also deserving of comment is the 54% of admissions on medical indication, as hypothetically following medical judgement, which could illustrate a crucial point in chronic disease management. Indeed, it could illustrate how in the comorbid elderly person well cared for by the caregiver, the functional crisis may be severe and require acute hospitalisation. In addition, it might be useful, with respect to limitations, to include a trend of acute beds in Switzerland, in order to better contextualise the beds associated with PAH.

Thanks

Author Response

Comment 1: The article is within the Journal aim and the topic is relevant. The text is clear and methods well described. 

Response 1: Thank you. 

 Comment 2: abstract: line 16 2009 to be revised as 1998

Response 2: The year was corrected.

 Comment 3: line 123-125 please remove

Response 3: The lines 123-125 were removed. 

 Comment 4: Discussion: Probably also deserving of comment is the 54% of admissions on medical indication, as hypothetically following medical judgement, which could illustrate a crucial point in chronic disease management. Indeed, it could illustrate how in the comorbid elderly person well cared for by the caregiver, the functional crisis may be severe and require acute hospitalisation.

Response 4: as the admissions were potentially avoidable, we believe that this indication might be due to the unavailability of the usual carer, the patient being hospitalized by a colleague who is not in charge of him/her. We added the following statement in the discussion:

“Slightly over half of all admissions were on medical indication. This suggests that patients with HF might not be able to contact their usual carer in due time, and that hospital admission is decided as a replacement for a consultation. Future studies should focus on the reasons for a carer to suggest a PAH for a patient with HF”

Comment 5: In addition, it might be useful, with respect to limitations, to include a trend of acute beds in Switzerland, in order to better contextualise the beds associated with PAH.

Response 5: we added the information in the results and we present the entire range of data in a new supplementary figure 2

“The total number of hospital days related to PAH for HF increased from 72,561 in 1999 to 148,390 in 2018 (supplementary figure 1), corresponding to an estimated number of 407 (minimum) to 478 (upper bound) occupied beds for the whole year, representing 1.8% of acute beds in hospital in 2018 which rose from 0.7% in 1999. (supplementary figure 2)

Reviewer 2 Report

Comments and Suggestions for Authors

There are some problems in statistical analyses, these should be resolved:

1. In the statistical analyses section, logistic regression analysis is mentioned. However, the table/s of logistic regression analysis results are not included in the article.

2. For some results, for example, line 149 says p<0.001, line 179 says p>0.05, line 181 says p<0.001. Which statistical tests were used to obtain these p values? A detailed explanation is required.

3. I could not access the supplementary files.

Author Response

Comment 1: In the statistical analyses section, logistic regression analysis is mentioned. However, the table/s of logistic regression analysis results are not included in the article.

Response 1: The tables of logistic regression analysis results are in the supplementary table 5; the results were briefly discussed in the results section (Line no: 181-186). 

Comment 2: For some results, for example, line 149 says p<0.001, line 179 says p>0.05, line 181 says p<0.001. Which statistical tests were used to obtain these p values? A detailed explanation is required.

Response 2: trends were assessed using linear regression, as indicated in the statistical methods, lines 109-110. We now provide the type of statistical method used lines 149 (now 153), 179 (now 175) and 181 (now 175)

Comment 3: I could not access the supplementary files.

Response 3: We have uploaded the supplementary again. Please see "Supplementary information_20241205.docx". 

Reviewer 3 Report

Comments and Suggestions for Authors

Phenomenon of potentially avoidable hospitalizations (PAH) for heart failure is very crucial for correct choose of directions of management of outpatient. So, the manuscript is relevant and can add some important information in understanding of phenotype of such patients.

The main idea of manuscript is to assess the trends in potentially avoidable hospitalizations (PAH) for chronic heart failure and to reveal factors, associated with this phenomenon among outpatients.
Certainly, this topic is very crucial, because heart failure is considered as one of the ambulatory care sensitive conditions, where better management in primary care could potentially avoid certain hospitalizations.
Heart failure remains one of the leading cause of hospitalizations and high mortality in the world, especially among elderly patients. So, the searching of significant predictors of potentially avoidable hospitalizations and their consequences is necessary for decreasing the burden of heart failure, because increasing portion of such admissions among all hospitalizations are extremely relevant and require particular attention.
The new sound of this study is characterized by providing additional risk factors for potentially avoidable hospitalizations, and not just hospitalizations due to heart failure outpatient. Authors confirmed that the number of potentially avoidable hospitalizations for heart failure has increased steadily and this situation is consistent with the current increase of HF cases in the general population. Mostly, the role of the general practitioners is believed to be linked to the increased amount of PAH.
The finding concerning balanced ratio between male and female patients in PAH and the peak amount of admissions for heart failure in elderly patients (the age of 80 and more) were revealed, which is consistent with other studies. Therefore, the importance of correct management of patients ages 80 years old is crucial and should be under precise consideration.
Minor comment for authors for the future perspectives: the information concerning the different phenotypes of heart failure due to left ventricular ejection fraction can find out additional predictors of PAH and improve structure management of outpatients.
Findings regarding the particularities of the patients with potentially avoidable hospitalizations (age, comorbidity) will improve the conclusion.
The references are relevant.

Author Response

Comment 1: Phenomenon of potentially avoidable hospitalizations (PAH) for heart failure is very crucial for correct choose of directions of management of outpatient. So, the manuscript is relevant and can add some important information in understanding of phenotype of such patients.

The main idea of manuscript is to assess the trends in potentially avoidable hospitalizations (PAH) for chronic heart failure and to reveal factors, associated with this phenomenon among outpatients.

Certainly, this topic is very crucial, because heart failure is considered as one of the ambulatory care sensitive conditions, where better management in primary care could potentially avoid certain hospitalizations.

Heart failure remains one of the leading cause of hospitalizations and high mortality in the world, especially among elderly patients. So, the searching of significant predictors of potentially avoidable hospitalizations and their consequences is necessary for decreasing the burden of heart failure, because increasing portion of such admissions among all hospitalizations are extremely relevant and require particular attention.

The new sound of this study is characterized by providing additional risk factors for potentially avoidable hospitalizations, and not just hospitalizations due to heart failure outpatient. Authors confirmed that the number of potentially avoidable hospitalizations for heart failure has increased steadily and this situation is consistent with the current increase of HF cases in the general population. Mostly, the role of the general practitioners is believed to be linked to the increased amount of PAH.

The finding concerning balanced ratio between male and female patients in PAH and the peak amount of admissions for heart failure in elderly patients (the age of 80 and more) were revealed, which is consistent with other studies. Therefore, the importance of correct management of patients ages 80 years old is crucial and should be under precise consideration.

Response 1: Thank you. 

 Comment 2: Minor comment for authors for the future perspectives: the information concerning the different phenotypes of heart failure due to left ventricular ejection fraction can find out additional predictors of PAH and improve structure management of outpatients.

Response 2: Thank you for your suggestion. The information regarding left ventricular ejection fraction is not available in the data for Switzerland obtained from the Swiss Federal Office for Statistics.

 Comment 3: Findings regarding the particularities of the patients with potentially avoidable hospitalizations (age, comorbidity) will improve the conclusion.

The references are relevant.

Response 3: We added the following in the conclusion.

"Patients with PAH for heart failure are mostly over ages 80 years old, with fewer comorbidities and admitted as emergency hospitalizations."

Reviewer 4 Report

Comments and Suggestions for Authors

Critical Review of the Article:

Title:
The title is clear and adequately describes the study's content. However, it could benefit from a more concise and engaging formulation, such as: "Trends in Avoidable Hospitalizations for Heart Failure in Switzerland (1998–2018): A Cross-Sectional Analysis."

Abstract:
The abstract includes the essential components: introduction, methodology, results, and conclusions. However, it lacks a critical analysis of the findings and does not explicitly mention key associated factors, such as comorbidities or high Charlson index scores.

Recommendation: Revise to include more critical details, such as the main limitations and practical implications. For instance, highlighting how these findings could guide primary care strategies would add value.

Introduction:

  • Strengths:
    The introduction effectively contextualizes the issue, emphasizing the importance of avoidable hospitalizations and the Swiss healthcare system, while citing relevant literature to support the study's focus.

  • Weaknesses:

    • The transition to the study's objectives feels abrupt.
    • From my clinical experience, including specific examples of how primary care in Switzerland may contribute to these hospitalizations would provide better context.

Recommendation: Improve transitions between paragraphs and tie the topic's relevance to practical experiences in primary care.

Methods:

  • Strengths:

    • The use of a nationally representative database and standardized definitions (OECD criteria) is commendable.
    • The multivariate analysis addressing factors like Charlson index is robust.
  • Weaknesses:

    • The exclusion of data with missing covariates is not sufficiently justified.
    • From a practical standpoint, it would be interesting to discuss whether the observed trends might have been influenced by changes in coding practices or health policies during the study period.

Recommendation: Elaborate on these limitations and consider their impact on the interpretation of the results.

Results:

  • Strengths:

    • Clear presentation of data using tables and graphs.
    • Comprehensive analysis of trends and determinants of avoidable hospitalizations.
  • Weaknesses:

    • Graphs lack interpretation within the text.
    • From my perspective as a physician, it would have been helpful for the results to highlight differences in admission patterns based on accessibility to primary care across different regions.

Recommendation: Summarize redundant data, enrich the interpretation of graphs, and emphasize clinically relevant patterns.

Discussion:

  • Strengths:

    • Strong connection between findings and previous literature.
    • Identification of key associated factors and trends.
  • Weaknesses:

    • The discussion does not sufficiently address how these findings could influence public health policies or daily clinical practice.
    • In my clinical experience, the role of primary care physicians should have been analyzed in greater depth, as they are key to reducing avoidable hospitalizations.

Recommendation: Expand the discussion on the practical impact of these findings, including specific intervention strategies.

Conclusions:
The conclusions are clear but too general. From a clinical practice perspective, it would be helpful to propose concrete actions for improving outpatient management.

Recommendation: Suggest specific measures, such as patient education programs for heart failure management or strategies to strengthen primary care.

Overall Strengths:

  1. Use of a robust national database and methodology.
  2. Relevant findings for Swiss public health policy.

Overall Weaknesses:

  1. Limited depth in analyzing clinical and policy implications.
  2. Underutilized graphs and tables for reinforcing conclusions.

Final Comment:

From my clinical experience, this article addresses an important topic and provides relevant data for medical practice and health policy. However, it requires improvements in the discussion and practical application of its findings. I recommend its publication after revisions that delve deeper into clinical implications and specific strategies for reducing avoidable hospitalizations.

Author Response

Comment 1: The title is clear and adequately describes the study's content. However, it could benefit from a more concise and engaging formulation, such as: "Trends in Avoidable Hospitalizations for Heart Failure in Switzerland (1998–2018): A Cross-Sectional Analysis."

Response 1: The title has now corrected into "Trends in Avoidable Hospitalizations for Heart Failure in Switzerland (1998–2018): A Cross-Sectional Analysis."

Comment 2: The abstract includes the essential components: introduction, methodology, results, and conclusions. However, it lacks a critical analysis of the findings and does not explicitly mention key associated factors, such as comorbidities or high Charlson index scores. Recommendation: Revise to include more critical details, such as the main limitations and practical implications. For instance, highlighting how these findings could guide primary care strategies would add value.

Response 2: We changed the abstract accordingly. Due to text limitations, we could not increase much further the size of the abstract

“Between 1999 and 2018, patients admitted with PAH for failure became older (% of patients aged over 80: 60.4% in 2018 vs. 49.2% in 1999), presented more frequently with a Charlson index <4 (65% vs 35%), were admitted more frequently as an emergency (89.0% vs. 60.7%), by the patient’s own initiative (31.5% vs. 13.9%), while ICU admission increased only slightly (8.6% vs. 7.6%) and length of stay decreased, median and [interquartile range] 8 [6-13] vs. 12 [8-18] days.”

Comment 3: Introduction:

  • Strengths:
  • The introduction effectively contextualizes the issue, emphasizing the importance of avoidable hospitalizations and the Swiss healthcare system, while citing relevant literature to support the study's focus.
  • Weaknesses:
    • The transition to the study's objectives feels abrupt.
    • From my clinical experience, including specific examples of how primary care in Switzerland may contribute to these hospitalizations would provide better context.

Recommendation: Improve transitions between paragraphs and tie the topic's relevance to practical experiences in primary care.

Response 3: we added more information regarding heart failure and made a specific paragraph. We also tried to improve the transitions between paragraphs. The text now reads:

“Potentially avoidable hospitalizations (PAH) are the hospitalizations that could have been avoided if there had been proper and effective outpatient primary care or early interventions outside the hospital settings. It has been widely used as an indicator of the performance, quality, and access to primary health care [ref]. Indeed, PAH can be useful to identify and prioritise areas to improve access to healthcare and efficiency of outpatient care, and consequently to reduce the disease burden. PAH for other ACSC such as asthma and chronic obstructive pulmonary disease in Switzerland showed an upward trend in the number of PAH in 2018 with almost double the number of PAH for COPD in 2008 [ref].

HF is a chronic and impairing disease, the worldwide prevalence of which has more than doubled between 1990 and 2019 [ref]. Despite being one of the countries with the best quality healthcare system in the world, there were considerable variations among PAH of HF in Switzerland in 2017 [ref]. However, no studies have been done regarding the trends and determinants of PAH for HF in Switzerland. Therefore, we aimed to analyse the trends, determinants, and consequences of PAH for HF in Switzerland for two decades from 1999 to 2018.”

Comment 4: Methods:

  • Strengths:
    • The use of a nationally representative database and standardized definitions (OECD criteria) is commendable.
    • The multivariate analysis addressing factors like Charlson index is robust.
  • Weaknesses:
    • The exclusion of data with missing covariates is not sufficiently justified.
    • From a practical standpoint, it would be interesting to discuss whether the observed trends might have been influenced by changes in coding practices or health policies during the study period.

Recommendation: Elaborate on these limitations and consider their impact on the interpretation of the results.

Response 4: the limitations are indicated in the corresponding chapter; we implemented it to read as follows:

Third, while the study covers a substantial time, it does not account for potential changes in healthcare policies, coding practices or reimbursement schemes that may have influenced hospitalization rates over the two decades. For example, increasing or decreasing use of HF code to modify reimbursements would artificially change the prevalence of the disease and modify the trends. Unfortunately, no information regarding any change in coding procedures could be found. Fourth, the exclusion of hospitalizations with missing covariates may introduce a selection bias, as only admissions with complete data (and possibly a better management) potentially impacting the generalizability of the findings regarding outcomes.

Comment 5: Results:

  • Strengths:
    • Clear presentation of data using tables and graphs.
    • Comprehensive analysis of trends and determinants of avoidable hospitalizations.
  • Weaknesses:
    • Graphs lack interpretation within the text.
    • From my perspective as a physician, it would have been helpful for the results to highlight differences in admission patterns based on accessibility to primary care across different regions.

Recommendation: Summarize redundant data, enrich the interpretation of graphs, and emphasize clinically relevant patterns.

Response 5: we increased the interpretation of graphs (legend). We also noticed that figure 2 was wrong and corrected it.

Comment 6: Discussion:

  • Strengths:
    • Strong connection between findings and previous literature.
    • Identification of key associated factors and trends.
  • Weaknesses:
    • The discussion does not sufficiently address how these findings could influence public health policies or daily clinical practice.
    • In my clinical experience, the role of primary care physicians should have been analyzed in greater depth, as they are key to reducing avoidable hospitalizations.

Recommendation: Expand the discussion on the practical impact of these findings, including specific intervention strategies.

Response 6: We added a new subsection 4.4 Implications for public health and clinical practice in the discussion.

“4.4. Implications for public health and clinical practice

PAH generally pose in significant economic burden and reduce the quality and efficiency of a healthcare system. Thus, reducing the number of PAH could significantly enhance the healthcare efficiency while also reducing the economic burden. Therefore, multidisciplinary strategies for public health and clinical practice are highly required. In terms of public health education, the European society of cardiology recommends education programs to patients with HF to identify the early symptoms of HF and to adopt good practices such as alternate daily weight monitoring and timely contacting their GP in the case of sudden body weight increases or in presence of early symptoms of HF [21]. As most PAHs were decided by doctors, it is also crucial to implement continuous medical education on the guidelines for the management of HF [22]. Other strategies such as fast track systems and improving primary care access for acute hospitalizations of HF, especially for the vulnerable and older patients, might also improve overall efficiency of healthcare system.”

Comment 7: Conclusions:

The conclusions are clear but too general. From a clinical practice perspective, it would be helpful to propose concrete actions for improving outpatient management.

Recommendation: Suggest specific measures, such as patient education programs for heart failure management or strategies to strengthen primary care.

Response 7: We added the following to the conclusion.

“The burden of PAH is highly significant, with important hospital resources that are mobilized, and healthcare policies such as patient education programs, enhanced primary care access and implication of updated guidelines to medical professionals, are required to introduce effective evidence-based health measures to reverse this trend, at the regional and national levels.”

Comment 8: Overall Strengths:

  1. Use of a robust national database and methodology.
  2. Relevant findings for Swiss public health policy.

Overall Weaknesses:

  1. Limited depth in analyzing clinical and policy implications.
  2. Underutilized graphs and tables for reinforcing conclusions.

Final Comment:

From my clinical experience, this article addresses an important topic and provides relevant data for medical practice and health policy. However, it requires improvements in the discussion and practical application of its findings. I recommend its publication after revisions that delve deeper into clinical implications and specific strategies for reducing avoidable hospitalizations.

Response 8: please see our reply to queries 6 and 7.

Round 2

Reviewer 2 Report

Comments and Suggestions for Authors

The requested corrections have been made.